# The Additional Value of Lower Respiratory Tract Sampling in the Diagnosis of COVID-19: A Real-Life Observational Study

**DOI:** 10.3390/diagnostics12102372

**Published:** 2022-09-29

**Authors:** Luca Morandi, Francesca Torsani, Giacomo Forini, Mario Tamburrini, Aldo Carnevale, Anna Pecorelli, Melchiore Giganti, Marco Piattella, Ippolito Guzzinati, Alberto Papi, Marco Contoli

**Affiliations:** 1Respiratory Medicine, Emergency Department, Azienda Ospedaliero Universitaria di Ferrara, 44124 Ferrara, Italy; 2Respiratory Medicine, Department of Translational Medicine, University of Ferrara, 44121 Ferrara, Italy; 3Radiology Department, Azienda Ospedaliero Universitaria di Ferrara, 44124 Ferrara, Italy; 4Section of Radiology, Department of Translational Medicine, University of Ferrara, 44121 Ferrara, Italy

**Keywords:** SARS-CoV-2, diagnosis, bronchoscopy, blind nasotracheal aspiration, chest CT

## Abstract

**Background:** Since December 2019, SARS-CoV-2 has been causing cases of severe pneumonia in China and has spread all over the world, putting great pressure on health systems. Nasopharyngeal swab (NPS) sensitivity is suboptimal. When the SARS-CoV-2 infection is suspected despite negative NPSs, other tests may help to rule out the infection. **Objectives:** To evaluate the yield of the lower respiratory tract (LRT) isolation of SARS-CoV-2. To evaluate the correlations between SARS-CoV-2 detection and clinical symptoms, and laboratory values and RSNA CT review scores in suspect patients after two negative NPSs. To assess the safety of bronchoscopy in this scenario. **Method:** A retrospective analysis of data from LRT sampling (blind nasotracheal aspiration or bronchial washing) for suspected COVID-19 after two negative NPS. Chest CT scans were reviewed by two radiologists using the RSNA imaging classification. **Results:** SARS-CoV-2 was detected in 14/99 patients (14.1%). A correlation was found between SARS-CoV2 detection on the LRT and the presence of a cough as well as with typical CT features. Typical CT resulted in 57.1% sensitivity, 80.8% accuracy and 92.3% NPV. Neither severe complications nor infections in the personnel were reported. **Conclusions:** In suspect cases after two negative swabs, CT scan revision can help to rule out COVID-19. In selected cases, with consistent CT features above all, LRT sampling can be of help in confirming COVID-19.

## 1. Introduction

In late December 2019, a cluster of pneumonia of unknown etiology was detected in the area of Wuhan, China. The agent responsible was identified as a coronavirus first named nCoV19, which was then changed to SARS-CoV-2 according to its genetic similarity with the etiologic virus of the severe acute respiratory syndrome in 2003 (SARS-CoV-1) [1]. Successively, the epidemic spread through all the continents until a pandemic was declared by the WHO on 11 March 2020 [2]. SARS-CoV-2 infection (COVID-19) has been shown to cause a wide spectrum of syndromes, varying from an unapparent, asymptomatic course and pauci-symptomatic infection mainly involving the upper respiratory tract and/or the gastrointestinal system to pneumonia and SARS requiring intensive care support, possibly leading to multiorgan failure and even death [3,4]. The diagnosis is based on molecular techniques for the identification of N and E genes of the SARS-CoV-2 genome using RT-PCR, and the nasopharyngeal swab (NPS) is still the recommended sample for the assay [5]. However, in clinical practice, the performance of the NPS RT-PCR is suboptimal, with a reported sensitivity of about 66% [6]. Several laboratory biomarkers, including a lymphocyte count, D-dimer assay and troponin I levels have been correlated to disease prognosis [4], but none of them provided a sufficient diagnostic yield [4,7,8,9]. Chest high-resolution computed tomography (CT) scans can reach very high sensitivity for the diagnosis of COVID-19 pneumonia [10]. Several radiological findings have been identified to support the diagnostic process, and some radiological societies have provided guidance in reporting chest CT abnormalities potentially attributable to COVID-19 pneumonia [11]. It is not that uncommon to experience cases of patients with clinical symptoms (mainly consisting of fever, cough and dyspnea) suggestive of COVID-19 but with nonconclusive radiological findings as well as negative NPS RT-PCR tests. Repetition of the test is highly recommended in these cases, given the epidemiological implications of allocating a SARS-CoV-2-infected patient to an inappropriate setting. Even a double negative result, however, might not be sufficient, and there is no consensus on how many samples can be considered enough to safely rule out COVID-19. In such cases, respiratory physicians may therefore be asked to help in the diagnostic process by sampling the lower airways. Indeed, lower respiratory tract (LRT) sampling through bronchial endoscopy with either bronchoalveolar lavage (BAL) or bronchial washing (BW or mini BAL) has proved useful in the etiologic definition of pneumonia, even in the case of viral pneumonia [12,13]. For the SARS-CoV-2 infection, a PCR assay performed on a BAL has shown very high sensitivity [6]. However, concerns have been raised concerning the procedure-related biological risk for the health care personnel (HCP) involved and the substantial cost for healthcare systems. In fact, in a statement, the American Association of Bronchology and Interventional Pulmonology (AABIP) discouraged the routine use of bronchoscopy in suspect cases [14]. Other associations released papers/recommendations with a similar position during the early phase of the epidemic [15]. 

Diagnostics have proven to be crucial to the COVID-19 pandemic response, and the issue of a prompt and correct diagnosis is still central and current in clinical practice [16]. Indeed, although the COVID-19 pandemic has led to a breathtaking development of vaccines, early treatments to prevent disease progression, especially in those who are most vulnerable, are now becoming available [17]. Moreover, postponing or modifying treatment schedules in certain subgroups, cancer or critical patients, for instance, in whom the clinical suspicion of COVID-19 is high but the test results are negative, may lead to worse outcomes [16,18].

Aiming to explore the utility of LRT sampling and its safety as a diagnostic tool in suspected SARS-CoV-2 infections, we designed a retrospective analysis of the data from patients undergoing lower respiratory tract (LRT) sampling because of suspected COVID-19 on the basis of clinical presentation and/or radiological findings but who were negative at NPS testing. 

## 2. Materials and Methods

### 2.1. Study Design, Population and Aims

This is an observational, retrospective, single center study. The analysis included all patients admitted to our university hospital between April 2020 and May 2021 with the following:-clinically suspected SARS-CoV-2 infection as per WHO recommendations (defined as respiratory symptoms of acute onset or contact with a confirmed case);-available chest CT scan;-at least 2 negative NPS RT-PCR tests for SARS-COV-2;-LRT sampling RT-PCR test for SARS-CoV-2.

Patients with only 1 negative NPS and patients without available CT scans were excluded. 

Aims of this study were to: (1) evaluate the yield of LRT sampling in this cohort of patients; (2) explore the relationship between LRT isolation of SARS-CoV-2 and clinical, laboratory and radiological characteristics; (3) assess the diagnostic value of internationally approved CT score in the diagnosis of NPS-negative but clinically suspect patients; (4) assess the safety of LRT sampling for both patients and HCP. 

### 2.2. Procedures and Analysis 

The LRT samplings were performed either by means of blind nasotracheal aspiration (BNTA) with a 14 Ch diameter suction catheter or bronchial endoscopy through bronchial washing (BW). Bronchoscopy was performed either with Pentax FB18V/FB15V fiberoptic bronchoscope (Pentax^®^, Tokyo, Japan) or a single use aScope™ 4 Broncho flexible bronchoscope (Ambu ^®^, Ballerup, Denmark). The decision as to which test to perform (BNTA vs. BW) was taken by the endoscopists in a multidisciplinary discussion with the referring clinician, based on clinical history review, clinical conditions/performance status and extent of the alterations on chest CT. The samplings were performed in a protected environment by expert personnel according to international recommendations [14,15,19,20,21]. SARS-CoV-2 detection on LRT samplings was retrospectively reviewed, and the threshold for positive results was set by our institutional laboratory at 40 or less RT-PCR cycles.

A three-dimensional score was set which aimed at revealing correlations between LRT test results (representing our diagnostic reference test) and chest CT as well as reported symptoms and laboratory biomarkers. The clinical score took the 3 most relevant symptoms (cough, dyspnea and fever) into account. Given the epidemic phase of the disease at the time of the study was carried out, close contact with a confirmed case was not considered necessary to suspect SARS-COV-2 infection in the presence of suggestive symptoms. In addition, included in the clinical score was respiratory failure, which is often present in COVID-19 patients on admission to hospital. Respiratory failure was defined as PaO2 <60 mmHg in ambient air via blood gas analysis or SpO2 <94% in ambient air. A score of 1 was assigned to presence and 0 to absence of each of the 4 items. 

Given the retrospective nature of the study, included in the laboratory score were variables considered clinically relevant after reviewing the literature and which were available for most of the patients included in the study: lymphocyte count, D-dimer assay, PCR and LDH. A 4-degree level score for lymphocyte count (0 = normal: >1500 cells/ µL; 1 = mild reduction: between 1000 and 1500/µL; 2 = moderate reduction: between 500 and 1000/µL; 3 = severe reduction: <500/µL) was arbitrarily set, and a 2-level score was set for the remaining variables (0 = normal, 1 = increased: >ULN for LDH, >500 ng/mL for D-dimer and >0.5 mg/dL for PCR). 

High-resolution CT scans were independently reviewed by two radiologists, blinded to clinical data in order to maintain objectivity in the identification of suspect imaging features. A score was assigned to each CT study according to the COVID-19 pneumonia imaging classification proposed by the Radiology Association of North America (RSNA): 0 = no pneumonia, 1 = atypical features for COVID-19, 2 = indeterminate features for COVID-19, 3 = typical features for COVID-19 pneumonia [11]. Cases raising disagreement between the two radiologists were reviewed and solved by consensus. Diagnostic performance was evaluated by considering as positive only CT scans with a score of 3. CT scans with scores of 0, 1 and 2 were considered as negative. Diagnostic accuracy was defined as the ratio between all cases in which CT score correctly assigned patients as either positive or negative for SARS-CoV-2 infection and the total number of patients. The complete scoring system adopted is summarized in Figure 1.

All the CT studies were acquired using the same scanner (LightSpeed64, GE Healthcare, Buckinghamshire, UK) in the supine position from lung apices to bases at full-suspended inspiration, with 1.25 mm section thicknesses, using standard acquisition parameters adjusted to the biometrics of patients. Images were reconstructed using a high spatial frequency kernel and visualized at window settings optimized for lung parenchyma (window level: −600 Hounsfield units (HU); window width: 1600 HU); a soft kernel was used for mediastinal window images (windows level: 50 HU; window width 350 HU). 

The study was approved by the local ethics committee, code: 825/2020/Oss/AOUFe. Informed consent for LRT sampling was obtained from all patients per ordinary clinical practice.

### 2.3. Statistics

All parametric and nonparametric variables are presented as mean and range. Statistical significance was set at 95% CI with *p* value < 0.05. Student’s *t*-test, chi-test and Mann–Whitney U test were used to compare characteristics between LRT-positive and -negative patients as appropriate. A binomial logistic regression analysis was performed to explore correlations between SARS-CoV-2 detection in LRT (dependent binary variable) and clinical, radiological, laboratory variables, the relative scores and combined score (independent variables). Included in the analysis were variables with significant correlation in univariate analysis and blood lymphocyte count as the laboratory variable with less missing data. Univariate analysis was performed using Graph Pad Prism © San Diego, CA (USA), version 7 and binomial logistic regression analysis using IBM ^®^ SPSS^®^, Armonk, NY (USA), version 27. 

## 3. Results

### 3.1. Study Population

Between April 2020 and May 2021, 122 patients underwent LRT sampling either through bronchoscopy or BNTA after at least two negative NPSs for suspect SARS-CoV-2 infection and were screened for inclusion in the study. Twenty-three patients were excluded because of the following reasons: 17 patients had undergone only one NPS before LRT sampling maneuvers, and five patients did not undergo a CT scan before the bronchoscopy. The remaining 99 patients were included in our analysis. At the time of LRT sampling, two patients were admitted to an ICU (critical respiratory condition), 17 to a pulmonology ward (intermediate respiratory condition) and the remaining 80 to an internal medicine or infectious disease ward (mild/intermediate respiratory condition). 

The characteristics of the study population (n = 99) are reported in Table 1. 

The mean age was 67.3 y, and most of the patients were male (69.7%). Fifty-eight patients (58.6%) had respiratory failure at presentation. All the patients had lung abnormalities in their CT scans. The mean time from the appearance of symptoms to CT scan was 4.93 days. Data were missing in three patients for the lymphocyte count, in 29 patients for the D-dimer, in 18 patients for the LDH, and in 6 patients for the PCR. 

LRT sampling was performed at a mean of 8.87 days after symptom onset. BNTA was performed in 21 patients (21.2%), while endoscopy was performed in 81 (81.8%); three patients underwent both types of sampling. SARS-CoV-2 was identified in LRT samples from 14 cases (14.1% of the included population). These comprised 11 BWs and three BNTAs. Negative tests were obtained in the remaining 85 patients. Three patients had been vaccinated at the time of testing: one with one dose, and two had received a second dose. 

### 3.2. Score Distribution 

Table 2 shows the score distribution in the overall population of the study. 

Only 11 patients were negative at the clinical evaluation (11.1%), while 5 patients showed a clinical score of 4 (5.0%). All patients presented with at least one laboratory abnormality, the most common being increased PCR. After revision by the two radiologists, 21 chest CT scans (21.2%) were scored as 3 (meaning consistent with COVID-19), 45 (45.5%) were scored as 2 (indeterminate for COVID-19), and 27 (27.3%) were scored as 1 (atypical for COVID-19), while the remaining 6 (6.1%) were scored as 0 (no signs of pneumonia). 

### 3.3. Chest CT Features and SARS-CoV-2 Detection

Table 3 summarizes the diagnostic performance of chest CT scans for the SARS-CoV-2 infection. 

Eight out of the twenty-one typical CT cases proved positive at LRT testing. The CT features were scored as 2 in four of the remaining six patients with positive LRT tests, and as 1 in the two remaining patients. Revision of the clinical data revealed that two of the patients with false negative CT were re-admitted after previous hospitalization for COVID-19 and had been discharged after clinical resolution and two consecutive negative NPSs as per local protocol. The reason for re-admission in both cases was the sudden appearance of respiratory symptoms and fever. The calculated diagnostic performance of the CT scan resulted in a sensitivity of 57.1%, a specificity of 84.7% and a positive predictive value and negative predictive value of 38.1% and 92.3%, respectively. Diagnostic accuracy was 80.8%.

### 3.4. SARS-CoV-2-Positive vs. -Negative Patients

The comparison between SARS-CoV-2-positive and -negative patients is shown in Table 4.

Positive cases were significantly younger, with no significant difference in gender distribution compared to negative patients. Among positive patients, a significantly higher percentage presented with a cough. No statistically significant difference in the prevalence of other reported symptoms was detected. No statistically significant difference was found in the laboratory values between SARS-CoV-2-positive and-negative patients. The mean time between the onset of symptoms and HRCT performance was found to be shorter among those with negative LRT sampling compared to SARS-CoV-2-positive patients (4.06 days vs. 8.85 days p 0.002), but no significant difference was observed between the time from onset of symptoms and LRT sampling in the two groups (8.35 days vs. 11.35 for negative and positive patients, respectively; p 0.08). 

The mean clinical score was not statistically different between SARS-CoV-2-positive and -negative patients (2.29 vs. 1.86, respectively). The mean laboratory scores (3.54 vs. 3.36 in the SARS-CoV-2-negative and -positive groups, respectively) showed no statistically significant difference. A statistically significant difference was, however, detected in the CT revision score, showing higher values in positive cases compared to the negative ones (mean 2.43 vs. 1.72, p 0.003). A statistically significant correlation was also detected between the total score (the sum of clinical, laboratory and radiological scores) and positivity for SARS-CoV-2 (mean 7.06 vs. 8.33 for negative and positive patients, respectively; p 0.036). 

Binomial logistic regression analysis (shown in Table 5) confirmed a correlation between SARS-CoV-2 infection detected in the LRT with age, the presence of a cough and the CT score, while it did not confirm a correlation with the total score.

### 3.5. Microbiological Analysis on LRT Sampling

LRT sampling resulted in the isolation of pathogens other than SARS-COV-2 in 42 out of 99 cases (42.4%). Microbiological analysis was not performed on 16 cases, mostly (13/16) consisting of BNTA due to less material being collected. The details of the isolated pathogens are provided in Table 6. It is noteworthy that in 26 (26.3%) cases, a mycotic infection was detected, with *Candida* spp. as the most common agent. Among bacteria, *Gram negative* spp. were the most frequently isolated, and *P. Aeruginosa* was the most consistently isolated. 

### 3.6. Complications

Eleven complications (11/122, 9.01%) were reported, mainly consisting of a drop in peripheral blood oxygen saturation during the procedure. In most cases, no specific intervention was required, with spontaneous resolution after the completion of the examination. In two cases, the drop in oxygen saturation prompted early termination of the bronchoscopy, and in one out of two, it required administration of high flow oxygen and IV steroids. During bronchoscopy, all patients with hypoxemia were already under an oxygen supply at the beginning of the procedure, and in the most severe case, pulmonary embolism was subsequently detected at a contrast-enhanced CT scan, accounting for higher procedure risks. Other mild complications included epistaxis and mucosal bleeding during the procedure, and one case of excessive sedation rapidly resolved with the administration of flumazenil. No complications were reported with the BNTA procedures. No site personnel were infected by SARS-CoV-2 as a consequence of the procedures.

## 4. Discussion 

This study aimed to: (1) assess the yield of LRT sampling in a cohort of patients clinically suspected of COVID-19 despite at least two negative NPSs; (2) explore the relationship between LRT isolation of SARS-CoV-2 and clinical, laboratory and radiological characteristics (3) assess the diagnostic value of an approved international CT score in the diagnosis of NPS-negative but clinically suspect patients; and (4) assess the safety of LRT sampling for both patients and HCP.

We reported 14 positive cases out of 99 (14.1% of the analyzed population). A correlation was only found between the positivity of LRT samples and presence of a cough but not with the other clinical or laboratory variables considered (lymphocyte count, D-dimer assay, LDH and PCR). Revision of the CT scans by two radiologists using the RSNA classification system showed a sensitivity of 57.1%, a diagnostic accuracy of 84.7% and a very high negative predictive value (92.3%). A correlation was found between typical CT features and the LRT isolation of SARS-CoV-2. A positive correlation was also revealed with our combined score, most likely driven by the cough and CT score. In fact, multivariate analysis confirmed all correlations except the one with the total score.

With the emergence of the SARS-CoV-2 pandemic, it became more and more obvious that NPSs could not be considered a perfect reference test for case identification, just as international guidelines could not recommend a specific number of NPSs to be performed in order to safely rule out COVID-19. On the other hand, international societies and expert panels considered bronchoscopy to be very hazardous for HCP and relatively discouraged its use for confirmation of the SARS-COV-2 infection [14,15].

Early studies exploring the role of bronchoscopy in COVID-19 analyzed heterogeneous populations with different SARS-CoV-2 prevalence, patient selection criteria and conclusions [22,23,24,25]. The initial reports from Italian groups mainly underlined concordance between negative NPSs and imaging (either CT or chest X-rays) and negative BALs for SARS-CoV-2 in clinically suspected COVID-19 [22,23]. Ora et al. also reported pre-bronchoscopy negative SARS-CoV-2 serology in their population, which is consistent with a lower pretest probability of COVID-19. Later multicenter studies confirmed a good overall correlation between negative NPSs and bronchoscopic lower respiratory tract samples but found a significant proportion of patients with positive BALs despite negative NPSs [24,25]. Of note, these studies reported a high overall prevalence of SARS-CoV-2 in BAL sampling (30 to 55%). These data were confirmed in a more recent international multicenter study analyzing bronchoscopy on suspected or confirmed COVID-19 patients for diagnostic or therapeutic reasons, showing a 44% detection of SARS-CoV-2 in the BAL fluid [26].

We reported a 14.1% prevalence of SARS-CoV-2 detection in LRT sampling in our population. Such a lower prevalence most likely reflects a difference in the epidemiologic scenario as well as a difference in patient selection; in fact, our analysis only included suspect patients with at least two consecutive negative NPSs in order to lower the false negative rate and not to overestimate the yield of bronchoscopy in view of the risks for HCP. Moreover, at the time of analysis, serology for SARS-CoV-2 was not available in our hospital setting. Hence, the conclusions of the work from Ora et al. do not apply to our cohort: by avoiding LRT sample analysis, 14 cases would have been missed, potentially increasing the risk of an in-hospital spread of the infection at a time of great pressure.

The potential of chest CT in the diagnosis of COVID-19 was first suggested by Ai et al. who calculated a sensitivity of 97% for SARS-CoV-2-associated pneumonia after reviewing more than 1000 CT scans from Chinese patients [10]. Since then, knowledge of the radiological appearance of SARS-CoV-2-related lung alterations has increased, leading to the development of standardized report systems [11,27]. A Cochrane meta-analysis by Salameh et al. including 78 studies evaluating the CT scans of 8105 participants found a 93.1% sensitivity for studies including confirmed cases and a 86.2% sensitivity in studies including suspect cases. However, specificity was found to be much lower (18.1%) [28]. Another meta-analysis by Kovacs et al. reported a sensitivity between 67 and 100% for chest CT. Interestingly, the authors noticed that the sensitivity of the NPS test clearly influenced the calculated CT scan specificity, which may therefore be underestimated [29]. Korevaar et al. retrospectively evaluated the added value to RT-PCR testing of chest CT scans analyzed with the CO-RADS score. The authors found a CT score consistent with COVID-19 in 92.9% of the patients with an initial positive RT-PCR test. They also reported that 13 out of 38 cases with positive CT scans but negative NPS tests ultimately had a confirmed SARS-CoV-2 infection after repetition of the RT-PCR testing. As in previous reports, specificity was lower, being only 25% [30]. A subsequent review of the CT scans in 46 patients from the study by Patrucco et al., using the recommended scores of international societies, showed good correlation between the suggestive features and BAL positivity for SARS-CoV-2 for both the CO-RADS and the RSNA classification system [31]. 

It is important to note that the diagnostic performance of CT scans does not depend merely on the prevalence of the disease in the population studied but is also affected by the nonspecific findings of COVID-19 that overlap with those of other conditions, resulting in a substantial rate of false–positive examinations [27]. Although accumulating data may suggest that a few radiological signs may be considered more specific for COVID-19 pneumonia [32,33], CT appearance is, indeed, not only nonspecific but also time-dependent, displaying predictable spatio–temporal changes [34,35]. Two main reasons could be given for false–negative results: first, symptomatic patients may not show lung abnormalities in the early course of the disease; second, a considerable number of patients with symptomatic upper respiratory tract infections do not develop pneumonia [36]. Another factor to be considered is the lack of a good quality reference standard for the diagnosis, as this standard will clearly influence the diagnostic performance of other tests [37]. Validated scores may help to increase the performance of chest CT: both RSNA or CORADS diagnostic classification systems may reduce observer variation and improve clinical communication [38]. 

Overall, our data support the role of chest CT in COVID-19 diagnosis, especially the use of a standardized scoring system. Indeed, the independent review of the CT scans by two radiologists resulted in good sensitivity, specificity, NPV and diagnostic accuracy. However, we found a lower sensitivity than previously reported. Several explanations may be put forward: first, by only including patients with at least two negative NPSs in our analysis, we lowered the incidence of infection in our population, leading to a higher specificity and NPV, which may not be applicable to other scenarios. The fact that suspicion of COVID-19 was raised on a clinical basis but not through a reproducible algorithm which may have included patients with a lower pre-test likelihood of COVID-19, is another factor that emerged. In addition, in order to not overestimate the CT performance, only cases showing typical features were considered as positive; therefore, we found a similar sensitivity as previously reported by Patrucco et al. in an analogous scenario, although in a smaller population (61% reported sensitivity for RSNA typical score) [31]. Finally, included in the analysis were two patients who had already been admitted and treated with a COVID-19 diagnosis and had been discharged from hospital with negative RT-PCR on NPS. It has already been described that BAL retains RT-PCR positivity for longer than NPS [39], though it is not clear whether this positivity really reflects infectivity. The CT features in these two cases may not represent the acute phase of the disease, and could, therefore, be misleading for the reading radiologists [40]. By excluding these two patients from the analysis, CT sensitivity would have been 66.7%, and diagnostic accuracy would have been 82.5%. Finally, revising the time from onset of symptoms and chest CT, we found false–negative CT scans had been performed significantly earlier in the course of the disease, therefore potentially misleading the reading radiologists. This raises the question of the optimal time to perform chest CT in the suspicion of COVID-19 in symptomatic patients, for which, at present, there is no consensus. In our scenario, having the CT scan revised and scored according to an objective and repeatable score is key to interpreting imaging features, as imaging revision by a respiratory clinician would have led to a greater number of false positives, given the presence of ground glass opacities in most of the cases.

Our results are consistent with published works on the topic, confirming the overall good concordance between negative CT findings, negative NPSs and negative LRT RT-PCR testing. Our study confirms that even negative NPSs cannot safely rule out the SARS-CoV-2 infection, especially when typical features are detected on CT scans, as reported by other groups [22,23,24,31], but at least two negative samples may significantly lower the number of examinations for diagnostic purpose, therefore protecting HCP as indicated by experts [21]. Our data also seem to suggest that as most symptoms are overlapping in different diseases, they should not be considered disease specific per se, but the presence of a cough should raise suspicions in the appropriate epidemiologic setting. We confirm that in these cases, most of the interest in laboratory values lies in their predictive value for clinical outcomes [4,7,9]. 

Cumbo-Nacheli et al. report a better performance of BAL than for BW [26]. Our work suggests the good performance of BW, as reported by other groups [24]. BW seemed, in the opinion of the clinicians involved in this study, more tolerable than BAL in a population with high respiratory complaints. We also included BNTA as a sampling technique for its less invasive nature and ease of performance, even for less trained personnel. In selected cases, BNTA can prove a valuable choice with lower risks of complications, considering the high prevalence of pre-test respiratory failure. 

Reported complications in the present study were at 9%; none of which were severe, and most consisted of a drop in oxygen saturation during the sample collection, similar to previously published series [25,26]. However, differences in the prevalence/severity of events are likely due to the different populations included (i.e., the prevalence of critically ill patients) and the subjective criteria of clinicians in the reporting of adverse events.

LRT sampling may also provide significant insight into differential diagnosis. Indeed, other groups found specific infective agents other than SARS-CoV-2 in a significant proportion of their patients [22,25,26,41,42,43]. In our cohort, BW identified respiratory infection by either bacteria or fungi in more than 40% of the included population, with overlapping bacterial and fungal infection in 13.6%. This confirms the high value of LRT sampling in managing lung infections, potentially leading to clinical changes in terms of antibiotic choice, with figures similar to those reported by Cumbo-Nacheli et al. (with altered clinical management in 48% of the patients) [26]. BNTA proved less effective for this purpose, mainly because less material was collected.

Our study certainly has all the limitations connected with its retrospective design, including the aforementioned selection biases. Other limitations include its monocentric nature and the relatively small number of cases analyzed. The diagnostic performance of a chest CT should be evaluated with regards to the low prevalence of infection and the epidemiological scenario of this population, which may change over time as new variants emerge and as a consequence of vaccination of (hopefully) the majority of the population worldwide. However, our study clearly reflects a real-life clinical scenario in a university center hub during the pandemic phase of the infection, where promptly confirming the SARS-CoV-2 infection is mandatory to avoid in-hospital spreading of the disease. 

To the best of our knowledge, we are the first to assess a repeatable combined clinical, laboratory and radiological score in the diagnostic workup of COVID-19, showing good correlations with a cough, CT scans and multidimensional scores. Our data suggest that COVID-19 is a multimodal diagnosis which should not only rely on a single test but rather on a panel of tests, including CT scans and LRT samples, especially when a CT scan is consistent with COVID-19 pneumonia, and less invasive tests cannot be conclusive. The role of bronchoscopy in the context of COVID-19 is still being debated as thoroughly presented in a recent review [44]. Multidisciplinary discussion, including at least radiologists and pulmonologists, is advisable in more complex cases. Our data also show that bronchoscopy can be performed while preserving HCP safety when protective equipment is used, as indicated in several reports and expert recommendations. It must be underlined that SARS-CoV-2 diagnosis is not only necessary in order to prevent the spread of the infection but also to access treatment (e.g., participation in clinical trials) and assure safety before undertaking immunosuppressive or chemotherapy treatment. Other applications of bronchoscopy in suspect cases can include differential diagnosis with other infectious diseases and the clearance of mucous secretions and plugs [26,44].

## 5. Conclusions

When clinical data, laboratory analysis and RT-PCR tests on NPSs are unable to lead to a definite confirmation or exclusion of a SARS-CoV-2 diagnosis, CT scan revision using validated scoring systems and lower respiratory tract sampling represents a useful tool. Bronchial washing (or mini BAL) can be effective in diagnosing SARS-CoV-2 and/or other respiratory infections. Multidisciplinary discussion involving at least a radiologist and a respiratory physician to evaluate the need for LRT sampling as well as the performance of bronchoscopy following international recommendations on protective equipment can minimize the potential risks for patients and HCP. 

## Figures and Tables

**Figure 1 diagnostics-12-02372-f001:**
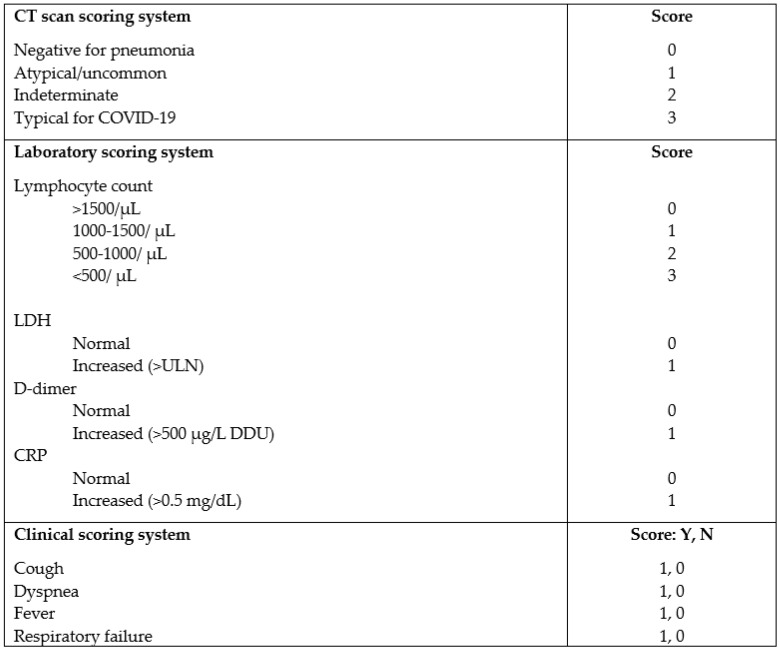
CT scan, laboratory and clinical scoring system. LDH=lactate dehydrogenase, CRP= C reactive protein, ULN= upper limit of normality, DDU = D-dimer unit, Y = yes, N = no.

**Table 1 diagnostics-12-02372-t001:** Main characteristics of patients with clinical suspicion of COVID-19 who underwent lower respiratory tract (LRT) sampling. # respiratory failure was defined as PaO2 <60 mmHg in ambient air or SpO2 <94% in ambient air. * includes: no alterations, emphysema, tree-in-bud, interstitial thickening with reticular pattern, pleural effusion. ** N.3 patients underwent both BNTA and BW. *** calculated including all screened bronchoscopies.

**Number of Patients**		N = 99
**Age**	years, mean (range)	67.3 (19–94)
**Gender**	n (%)	
Male		69 (69.7%)
Female		30 (30.3%)
**Presenting symptoms**	n (%)	
Fever		54/99(54.5%)
Cough		27/99 (27.3%)
Dyspnea		51/99 (51.5%)
Respiratory failure #		58/99 (58.6%)
**Mean time from symptoms to HRCT**	days, mean (range)	4.93 (1–24)
**Mean time from symptoms to LRT sampling**	days, mean (range)	8.87 (2–28)
**Radiological abnormalities at HRCT**	n (%)	
Main abnormal findings		
Unilateral ground glass		7/99 (7.1%)
Bilateral ground glass		40/99 (40.4%)
Ground glass and consolidation		31/99 (31.3%)
Consolidation w/o ground glass		15/99 (15.1%)
Other *		6/99 (6.1%)
**LAB values**		
Lymphocyte count	cells/µL, mean (range)	1492.2 (70–15800)
LDH	U/L, mean (range)	337.5 (121–4428)
D-dimer	µg/L DDU, mean (range)	1534.2 (130–17500)
CRP	mg/dL, mean (range)	9.9 (0.02–36.38)
**LRT sampling technique**	n (%)	
Bronchial washing (BW)		81/99 (81.8%)
Blind nasotracheal aspiration (BNTA)		21/99 (21.2%) **
**LRT sampling result**		
SARS-CoV-2 detection	n (%)	14/99 (14.1%)
Bronchial washing	n (%)	11/81 (13.6%)
Bronchial aspiration	n (%)	3/21 (14.3%)
**Complications**	n (%)	11/122 *** (9.1%)

**Table 2 diagnostics-12-02372-t002:** Clinical, LAB and radiological score distribution in the study population.

**Clinical Score**	n (%)	
0		11/99 (11.1%)
1		22/99 (22.2%)
2		35/99 (35.4%)
3		26/99 (26.3%)
4		5/99 (5.0%)
**LAB score**	n (%)	
0		0/99 (0.0%)
1		8/99 (8.1%)
2		11/99 (11.1%)
3		18/99 (18.2%)
4		15/99 (15.1%)
5		14/99 (14.1%)
6		2/99 (2.0%)
Missing data		31/99 (31.3%)
**CT revision score**	n (%)	
0		6/99 (6.1%)
1		27/99 (27.3%)
2		45/99 (45.5%)
3		21/99 (21.2%)

**Table 3 diagnostics-12-02372-t003:** CT scan performance in identifying SARS-CoV-2 infection. PPV=positive predictive value, NPV= negative predictive value.

	N = 99
**Sensitivity**	57.1% (8/14)
**Specificity**	84.7% (72/85)
**PPV**	38.1% (8/21)
**NPV**	92.3% (72/78)
**Diagnostic accuracy**	80.8% (80/99)

**Table 4 diagnostics-12-02372-t004:** Comparison between positive and negative samples.

Negative LRT Samples	N = 85	Positive LRT Samples N = 14		*p* Value
**Age:** mean, years (range)				
70.1 (19–89)		50.5 (19–94)		<0.0001
**Gender:** n (%)				
Male 59 (69.4%)		Male 10 (71.4%)		>0.99
Female 26 (30.6%)		Female 4 (28.6%)		
**Clinical symptoms prevalence:** n (%)				
Fever	44 (51.8%)	Fever	10 (71.4%)	0.24
Cough	19 (22.4%)	Cough	8 (57.1%)	0.01
Dyspnea	44 (51.8%)	Dyspnea	7 (50%)	>0.99
Respiratory failure	51 (60.0%)	Respiratory failure	7 (50%)	0.56
**LAB values:** mean (range)				
Lymphocyte count (cells/µL)	1565.5	Lymphocyte count (cells/µL)	1063.0	0.34
(70–15,800)§		(210–1630)		
LDH (U/L)	343.5 (121–4428)	LDH (U/L)	306.5 (165–532)	0.80
D-dimer (µg/L DDU)	1976.1 (130–17,500)	D-dimer (µg/L DDU)	887.1 (155–3380)	0.35
CRP (mg/dL)	10.2 (0.02–36.38)	CRP (mg/dL)	8.18 (1.23–21.14)	0.4
**Time from symptoms to HRCT: days, mean (range)**				
4.06 (1–24)		8.85 (1–19)		0.002
**Time from symptoms to LRT sampling: days, mean (range)**				
8.35 (2–28)		11.38 (2–23)		0.08
**Clinical score: mean (range)**				
1.86 (0–4)		2.29 (0–4)		0.21
**LAB score: mean (range)**				
3.35 (1–6)		3.36 (2–4)		0.59
**CT revision score: mean (range)**				
1.72 (0–3)		2.43 (1–3)		0.003
**TOTAL score: mean (range)**				
7.06 (2–12)		8.33 (4–10)		0.036

**Table 5 diagnostics-12-02372-t005:** Binomial logistic regression analysis of SARS-CoV-2 detection in LRT.

Dependent Variable	Independent Variables	B Coefficient	S.E.	Wald Test	*p*	OR
LRT detection of SARS-CoV-2	Age	−0.125	0.049	6.613	0.010	0.883
	Cough	3.872	1.659	5.451	0.020	48.051
	Lymphocytes	−0.001	0.001	1.133	0.287	0.999
	CT scan score	1.916	0.937	4.179	0.041	6.795
	Total score	−0.175	0.518	0.114	0.429	0.839

**Table 6 diagnostics-12-02372-t006:** Isolated germs from bronchial washing. * 13/16 coming from BNTA.

**Bronchial Washing Cultural Examination**	**n (%)**	N = 99
Not available/not executed		16/99 (16.2%) *
Negative		41/99 (41.4%)
Positive		42/99 (42.4%)
Bacterial infection		16/99 (16.2%)
Mycotic infection		15/99 (15.2%)
Combined bacterial and mycotic		11/99 (11.1%)
*Gram negative* spp.		21/99 (21.2%)
*Gram positive* spp.		4/99 (4.0%)
*Mycobacteria*		2/99 (2.0%)
*Candida* spp.		24/99 (24.2%)
*Aspergillus* spp.		1/99 (1.0%)

## Data Availability

The raw data used/analyzed in the current study are available upon reasonable request to the corresponding author and ethics committee (CE-AVEC).

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
