# Peer review of "The Additional Value of Lower Respiratory Tract Sampling in the Diagnosis of COVID-19: A Real-Life Observational Study"

_diagnostics, 2022, doi:10.3390/diagnostics12102372_

Round 1
Reviewer 1 Report (Previous Reviewer 1)
The authors tried to describe the additional value of lower respiratory tract sampling in the 2 diagnosis of COVID-19. A real life observational study. The authors have edited the manuscript well and is now of good quality.
Reviewer 2 Report (Previous Reviewer 2)
The authors have made the corrections indicated by me previously and enriched the text with other contributions.
This manuscript is a resubmission of an earlier submission. The following is a list of the peer review reports and author responses from that submission.
Round 1
Reviewer 1 Report
The authors try to report The additional value of lower respiratory tract sampling in the diagnosis of COVID-19. A real life observational study.
The topic could be change to The authors try to report The additional value of lower respiratory tract sampling in the diagnosis of COVID-19, a real life observational study.
Although the manuscript is well written please re-edit it as it has several grammatical and punctuation problems. For example, in “line 14” please change spread to spreading, “line 16” use “of” after yield, line 23 use “the” before presence, “line” 24 staff’s should be change to staff, line 75, and so on.
The references are not sufficient for an original article.
Use reference for line 71.
It is highly recommended to Rearrange the tables.
Reviewer 2 Report
As a clinician in a ward with patients with COVID-19 since March 2020, I know the importance of confirming the diagnosis of COVID-19, in order to treat the patient and avoid nosocomial transmission.
The authors analyze the use of CT for the diagnosis of COVID-19 in patients with two previous NPS negative tests, corroborating the existing literature on the subject. They also analyze the usefulness and safety of bronchoscopy for LRT sample collection in these patients.
The abstract is clear and concise, it exposes the most important results of the study and its implications.
The introduction is well thought out, it places the objective in its current clinical context.
In material and methods, the authors should specify the diagnostic test used, since the performance of the PCR since April 2020 has improved significantly. Authors should also specify the type of multivariate analysis performed, since they analyze qualitative and quantitative variables.
The results are detailed and clear. However, the presentation of Table 3 is confusing. Table 5 should include the data and the risk analysis, not just the p result of the multivariate analysis.
The discussion is simple, based on the results, and makes an adequate review of the available bibliography, contextualizing the findings. The limitations of the study are exposed by the authors. Perhaps should be mentioned that one of the main impediments to the use of bronchoscopy for the diagnosis of COVID-19 is its cost both in time and in personnel and resources. However, it is true that it seems to be useful in situations such as those presented by this study. Also, the authors could place more emphasis on its usefulness in the differential diagnosis with other infections.
The conclusions are clear, and are supported by the results obtained in the study.
Minor corrections
page 8 lines 191-192: "mean time between symptoms onset and HRCT performance resulted slightly shorter among those with negative LRT sampling as compared to SARS-CoV-2 positive patients". The difference was statistically significative and it almost doubled the days, it is not "slightly".
Reviewer 3 Report
The authors present a very interesting retrospective study. The structure, the idea and the presentation of their results is very nice providing evidence to the question they pose concerning the use of lower respiratory tract sampling. I think their study offers new data in the clinical practice of the physicians. My only concern is that we have seen that RT-PCR in BAL remains positive longer than in nasopharyngeal testing, fact that may reflect colonization and not clearly infectivity. However, the authors dispute this in the discussion section.